# Negative Bronchoscopy or Computed Tomography Radiation in Children with Suspected Foreign Body Aspiration? Pros and Cons

**DOI:** 10.3390/tomography11020017

**Published:** 2025-02-14

**Authors:** Mehmet Emin Çelikkaya, Ahmet Atıcı, İnan Korkmaz, Mehmet Karadağ, Çiğdem El, Bülent Akçora

**Affiliations:** 1Pediatric Surgery Department, Tayfur Ata Sokmen Medicine Faculty, Hatay Mustafa Kemal University, Hatay 31100, Turkey; ahmetatici06@gmail.com (A.A.); bakcora@hotmail.com (B.A.); 2Radiology Department, Tayfur Ata Sokmen Medicine Faculty, Hatay Mustafa Kemal University, Hatay 31100, Turkey; drinankorkmaz@gmail.com; 3Biostatistics Department, Tayfur Ata Sokmen Medicine Faculty, Hatay Mustafa Kemal University, Hatay 31100, Turkey; mkarad@gmail.com; 4Children’s Health and Diseases Department, Tayfur Ata Sokmen Medicine Faculty, Hatay Mustafa Kemal University, Hatay 31100, Turkey; cigdem.el@hotmail.com

**Keywords:** foreign body aspiration, negative bronchoscopy, low-dose computed tomography, radiation

## Abstract

Background: The aim of this study was to analyse patients who underwent bronchoscopy for suspected foreign body aspiration (FBA) and to evaluate the properties of computed tomography (CT) in preventing unnecessary bronchoscopy, which carries the risk of serious complications. Methods: All patients younger than 18 years of age who were evaluated for FBA at a tertiary children’s hospital between June 2014 and February 2023. Results: A total of 165 children were included in this study. In Group I (only bronchoscopy), the detection rate of FBA was 77.9%, whereas in Group II (CT ± bronchoscopy), the detection rate of FBA was 93%. Additionally, the negative diagnosis rate in Group II (7%) was significantly higher compared to Group I (22.1%). Conclusions: Low-dose chest CT is a highly effective and reliable imaging method for the diagnosis of FBA due to its rapid performance, minimal radiation exposure, and high sensitivity and specificity; therefore, it can prevent unnecessary bronchoscopies in suspicious cases and increase the positive bronchoscopy rate.

## 1. Introduction

Foreign body aspiration (FBA) remains an important cause of morbidity and mortality in childhood. It has been said to be the main cause of accidental deaths in the first year of life [1]. It is most commonly seen in children under 4 years of age, and hypoxic brain damage and death can be observed if intervened late [2]. The main feature in clinical diagnosis is the presence of an expulsive cough and laryngeal spasm as a respiratory defence reflex to aspiration of a foreign body. This clinical picture is called penetration syndrome. Penetration syndrome is characterised by cyanosis and asphyxia associated with coughing fits, but in 12 to 25% of cases, the clinic may be silent. The most common clinical signs in the acute phase are wheezing, localised reduction or loss of vesicular breath sounds, and intercostal retractions. In later periods, if the penetration syndrome is missed, the child frequently presents with a history of recurrent pneumonia in the same region [3]. It is very important to diagnose and remove the foreign body in the early period to reduce the incidence of mortality and postoperative complications.

Significantly abnormal physical examination findings and positive findings on direct radiography (presence of a radiopaque foreign body, unilateral hyperexpansion, or unilateral atelectasis) confirm that the diagnosis and bronchoscopy should be performed in these patients [4]. Unfortunately, the clinical picture is often unclear, and the clinician must decide which patients should undergo bronchoscopic evaluation [5]. Although direct radiographs are routinely used, they cannot exclude the diagnosis in the absence of positive findings because approximately one third of patients with FBA have a normal radiography [6]. Algorithms and multivariate models using anamnesis, physical examination, and plain radiographs still have only about 70% sensitivity and 60% specificity [7,8]. Rigid bronchoscopy, the gold standard for diagnosis and definitive treatment, remains an invasive procedure that requires exposure to anaesthesia in a patient with respiratory symptoms and carries the risk of exacerbation of reactive airway disease [5]. During bronchoscopy, complications ranging from transient desaturation to cardiac arrest occur at a rate of 2.6–14%, and mortality at a rate of 0.42–0.8%. In addition, in patients who underwent bronchoscopy with suspicion of FBA, the rate of confirmation of this diagnosis by bronchoscopy varies between 30% and 93% in the literature [5,8,9,10]. Currently, the standard technique for bronchial foreign body removal in children is rigid bronchoscopy under general anaesthesia, but this technique is used very frequently for both diagnostic and therapeutic purposes, resulting in a negative bronchoscopy rate of 10% to 61% [11,12].

Up to 40% of children may have no physical findings that would suggest a foreign body, and the children with foreign bodies could have normal radiography. Small foreign bodies can remain in the respiratory tract for weeks without symptoms, and the development of pneumonia may be the first sign of aspiration. If the air can freely pass the foreign body, especially in the early times after aspiration, auscultatory changes may not be detected. Although radiography is the main tool used to detect the presence of foreign bodies in the respiratory tract, it is insufficient for diagnosis, especially in the early stages, because time is required for atelectasis or emphysema to be observed [13].

Recent developments in multidetector computed tomography (CT) have shortened the acquisition time and improved image quality. Since the acquisition time is only a few seconds in a co-operative patient, it can be performed in children without sedation. According to various studies, the bronchial sensitivity of multidetector CT in the diagnosis of FBA is close to 100%, and the specificity is between 66.7% and 100%. False positive results are usually related to the presence of a mucus plug or artefact. No false negative results have been reported, but the sensitivity of this examination cannot be reliably determined because of the small sample sizes of the published series [3]. In addition, CT may provide the surgeon with precise information about the location and size of the bronchial foreign body, thus reducing the operative time in the patient undergoing rigid bronchoscopy. CT can also show associated lung lesions (emphysema, atelectasis, pneumothorax, and bronchiectasis) [3,14]. Therefore, there is a need for a diagnostic tool that is both superior to plain films and less invasive than bronchoscopy; this gap can potentially be filled by CT. The aim of this study was to retrospectively analyse patients who underwent bronchoscopy for suspected foreign body aspiration and to evaluate the properties of CT in preventing unnecessary bronchoscopy, which carries the risk of serious complications.

## 2. Material and Methods

All patients younger than 18 years of age who were evaluated for foreign body aspiration at a tertiary children’s hospital between June 2014 and February 2023 were included in the retrospective review. Ethics committee permission dated 1 September 2022 and numbered 32 was obtained. Patients were evaluated according to age (Figure 1), gender (Figure 2), and aspiration material. The patients were divided into two groups. Group 1 consisted of patients who underwent bronchoscopy only. Group 2 patients consisted of patients who underwent CT and then bronchoscopy. All patients presented with a history of suspected aspiration or were referred for bronchoscopy by a paediatrician due to a history of frequent and recurrent pneumonia. Direct radiography was taken in all patients. Low-dose non-contrast computed tomography was performed on patients who were unremarkable on direct radiography but had a history of likely aspiration. Bronchoscopy was performed in patients with a foreign body seen on plain radiographs, patients with no foreign body seen on plain radiographs but a foreign body detected on CT, and patients with clinical complaints and a history of suspected aspiration regardless of radiological appearance. Images of all patients who underwent CT were evaluated by the same radiologist.

All indicated patients underwent rigid bronchoscopy (Storz, Tuttlingen, Germany) under general anaesthesia (inner diameter of 3.5 mm and length of 30 cm). Foreign bodies were removed with rigid grasping forceps. Patients who had problems waking up from anaesthesia postoperatively were followed up in the intensive care unit; written informed consent was obtained from the parents of each paediatric patient before bronchoscopy was performed.

### 2.1. Screening Protocol

CT scans were performed on a 64-slice Toshiba Aquilion unit (Toshiba Medical System Corporation, Otawara-Shi, Japan, Model TSX101A, 5 mm slice thickness). Axial images were acquired in a supine position (feet first) from the lung’s apex to the costophrenic angle. CT scans were performed at low dose (30 mA, 80 kV) for the paediatric age group and without the use of any contrast agent. A lung window image sequence with a slice thickness of 1.25 mm was used for measurements. The coronal and sagittal multiplanar reformation was performed at 3 mm thickness. The data set was transferred and stored in DICOM format on the hospital PACS system. Parents with crying or moving children wore lead aprons to keep the child still during the CT scan. However, if intense artefacts appeared in the screen images of moving children, they were excluded from this study.

### 2.2. Anaesthesia Protocol

Anaesthesia was initiated and maintained with oxygen/nitrous oxide or sevoflurane, and a muscle relaxant was administered. The stomach was emptied before the procedure with a suction catheter. Rigid bronchoscopy was inserted after muscle relaxation. Patients were ventilated with manual bag ventilation through a rigid bronchoscope during the bronchoscopic procedure.

### 2.3. Hospitalisation Criteria and Follow-Up

Patients with a history of likely foreign body aspiration (had a history or witness of aspiration) who had mild/no clinical complaints and no foreign body was observed on radiological imaging were hospitalised for 48 h of observation. After discharge, the children were called for a check-up 2 weeks later.

### 2.4. Statistical Analysis

Categorical variables were analysed using Pearson chi-square tests with 2 × 2 tables. Sensitivity, specificity, and accuracy analyses were performed to assess the potential diagnostic ability of both CT and X-ray on FBA. Patients were considered to be true negatives if no foreign body was found on bronchoscopy. In addition, kappa statistics were used to determine the between CT and X-ray agreement for the FBA diagnosis. The kappa statistic was interpreted as follows: less than 0.00, poor agreement; 0.00–0.20, slight agreement; 0.21–0.40, fair agreement; 0.41–0.60, moderate agreement; 0.61–0.80, substantial agreement; and 0.81–1.00, almost perfect agreement. SPSS for Windows (version 25.0; SPSS, Inc., Chicago, IL, USA) was used for analysis. A *p* value less than 0.05 was considered significant.

## 3. Results

A total of 165 children who underwent bronchoscopy were included in this study. It was observed that 59.4% (n = 98) of the cases were girls and 40.6% (n = 67) were boys. The median age of the cases was 2 years, ranging from 0.5 to 18 years, and the interquartile range (Q3–Q1) value was 2. Direct radiography was performed in 100% of the cases (n = 165), and the CT scan was performed in 26.1% (n = 43). Other characteristics of the cases are presented in Table 1.

The foreign bodies removed from the children are shown in Table 2. According to Table 2, the most common aspirated foreign body in children was peanut 29.2% (n = 40), followed by kernel 13.14% (n = 18) and needle 11.67% (n = 16). Moreover, 47 patients had positive findings on direct radiography. In 24 patients, aspirated opaque substances (needle, stone, pen tip, metal, and tack) could be seen (Figure 1). In 23 patients, although no foreign body was seen in direct radiography, findings such as increased ventilation and atelectasis were observed (Figure 2). In 28 patients, no abnormality was detected on radiography, but foreign bodies were observed on CT (Figure 3). In one patient, two needles were observed at the same time; one was aspirated and the other was swallowed. In 2 patients, bronchoscopy was performed twice, one 2 months and one 4 months apart, because of repeated foreign body aspiration.

The presence of a foreign body detected by bronchoscopy was considered the gold standard, and the predictive values of CT and radiography devices in detecting foreign bodies and their compatibility with each other were investigated (Table 3).

The sensitivity and specificity values of the CT for detecting the presence of a foreign body were 100%, 100%, and 100%, respectively, and the total accuracy rate was (40 + 3)/43 = 100%. The compatibility of the CT with the bronchoscopy was determined to be at a statistically significant 1 (95% CI: 1–1; *p* < 0.001) substantial agreement level. When the results were analysed, it was determined that the sensitivity of the radiography device in detecting the presence of a foreign body was 34.1%, the specificity was 96.7%, and the total accuracy was (46 + 29)/165 = 45.4%. The agreement between radiography and the bronchoscopy device was at a statistically significant 0.144 (95% CI: 0.10–0.26; *p* = 0.001) at a slight agreement level.

The agreement between radiography and CT devices used to detect the presence of foreign bodies in children was analysed (Table 4).

According to Table 4, it was determined that the compatibility between radiography and CT devices was not statistically significant.

It was determined that there were statistically significant differences between groups in detecting FBA (*p* = 0.027). In Group II (CT ± bronchoscopy), the detection rate of FBA was 93%, whereas in Group I (only bronchoscopy), the detection rate of FBA was 77.9%; this difference was statistically significant. Additionally, it was observed that the negative diagnosis rate in Group II was significantly higher compared to Group I (Table 4).

## 4. Discussion

Foreign body aspiration in children is a potentially serious household accident that is a frequent presenting complaint to paediatric emergency departments [15]. More than 80% of foreign body aspiration cases occur in early childhood, and the highest incidence is observed between 10 and 24 months. The absence of molars and premolars and the tendency to bring all objects to the mouth explain the special predisposition of children in this age group [16,17]. Symptoms may vary significantly according to the location of the foreign body in the airways. When the foreign body is trapped in the larynx or trachea, respiratory distress or stridor immediately suggests the diagnosis. However, in the vast majority of cases (75 to 94 per cent), the foreign body migrates into the bronchi, and clinical symptoms are much less persistent [11,18]. Since the risk of complications related to the presence of a bronchial foreign body increases with the passage of time, it is important to make the diagnosis as soon as possible [3]. Currently, the standard technique for the treatment of foreign body aspiration in children is rigid bronchoscopy under general anaesthesia, but this technique is used very frequently for both diagnostic and therapeutic purposes, resulting in a negative bronchoscopy rate of 10% to 61% [3,19]. Most authors have stated that rigid bronchoscopy is the standard technique for the removal of bronchial foreign bodies in children with a success rate of more than 97% [6,16,20]. In our study, rigid bronchoscopy was negative in 30 patients among 165 patients. The rate of negative rigid bronchoscopy was 18.1%. All of these patients were in the non-CT patient population.

According to various authors, the complication rate related to rigid bronchoscopy varies between 2% and 22%. The most common complications are laryngeal oedema and pneumothorax, but more serious complications such as tracheal tear, bronchial tear, hypoxia, and cardiorespiratory arrest may also occur. Fortunately, these complications are rare [3,6]. Because of these rare but serious complications, it is important to reduce the rate of negative rigid bronchoscopy [3]. The skills and clinical experience of the surgeon and anaesthetist are also very important for the bronchoscopy procedure. Anaesthetists and surgeons should be aware of the severity of complications and have a plan to manage postoperative complications [21]. In our patients, intra-bronchial haemorrhage occurred in 5 patients, low oxygen saturation and bronchospasm in 8 patients, bradycardia in 2 patients, and cardiac arrest in 1 patient who returned to sinus rhythm with cardiac massage. No bronchoscopy-related mortality was observed in our patients.

According to Silva et al., the sensitivity and specificity of chest radiography increase when chest radiography is performed 24 h after aspiration [22]. In the acute phase, the sensitivity and specificity of chest radiography for the diagnosis of bronchial foreign body is low [23], and it is reported to be normal in 14 to 37% of cases in many studies [12,19,24]. In our study, 46 (34.1%) of 135 patients who had detected FBA on bronchoscopy had positive findings on DG. In 89 (65.9%) patients in whom FBA was detected in bronchoscopy, no finding in favour of FBA was observed on direct radiography.

Hegde et al. reported that CT was superior to plain films in the detection of foreign bodies in the respiratory tract [25]. Behera et al. evaluated patients who underwent CT and bronchoscopy for FBA and found that 59/60 of them were confirmed. The remaining case, which was suspicious on CT, was found to be a thick mucus plug by rigid bronchoscopy [26]. Gibbons et al. compared 64 patients who underwent bronchoscopy only with 69 patients who underwent CT and bronchoscopy and reported that the diagnosis of foreign body was excluded in 49 patients with CT, and unnecessary bronchoscopy was prevented [5]. In our study, 165 paediatric patients who underwent bronchoscopy were evaluated, and no foreign body was observed in 30 patients. In a group of 122 patients who underwent bronchoscopy alone, no foreign body was observed in 27 patients (22%), whereas in a group of 43 patients who underwent CT, no foreign body was observed in bronchoscopy performed because of the anamnesis and intense dyspnoea in the patients, although CT was negative in 3 patients (7%). In our study, the rate of negative bronchoscopy decreased from 22% to 7% in the CT group compared to the bronchoscopy-only group.

Qiu et al., in a 7-year study evaluating 48 patients who aspirated FB and had false negativity on CT, stated that 43.9% of the aspirated material consisted of sheet or dust. It was also stated that in more than 50% of the patients, the diagnosis was made at least 2 weeks later, and it took up to 2 years at the longest. It has also been stated that the image may be confused with pneumonia due to lung infiltrates in these patients [27]. In our study, the application period of the patients was between the same day and one month, and the number of patients with lung infiltration was only 2 (1.2%).

Kim et al. stated in their study that in addition to the use of CT preventing unnecessary bronchoscopies, the use of CT for diagnosis in suspicious cases can potentially prevent missed diagnosis in clinically suspicious cases, which can occur in up to 20% of cases with plain films [2]. Çiftçi et al. stated that up to 33% of FBA cases can be misdiagnosed as pneumonia based on clinical findings alone [6]. In our study, it was observed that 2 of 3 patients with negative FB on bronchoscopy had an asthma attack, and one patient had pneumonia. Another potential benefit of using CT to diagnose FB is its applicability for triage in public hospitals where a paediatric surgeon is not available. In these hospitals, unnecessary transfer to the referral hospital can be prevented in case of CT negativity [5]. In our study, 12 patients first applied to an external centre and were referred to us upon detection of FB on CT.

A CT scan is performed very quickly, usually within a few seconds. So, the children scarcely need any kind of sedation to get a high enough quality image in order to evaluate for a foreign body aspiration [5]. But there are the practical challenges associated with performing CT scans, such as the availability of expert radiologists during evening and night shifts, time constraints, cost implications, and the challenges of managing paediatric patients who may be uncooperative.

The use of fluoroscopy in the diagnosis of FBA is very limited in children; it can only be useful in demonstrating mediastinal displacement or decreased diaphragmatic movements on the side of aspiration. Fluoroscopy is a dynamic study that shows the ventilation of the lungs. However, it is difficult to ensure cooperation during inspiration and expiration in children. It is also a user-dependent examination, and false-negative rates reach up to 50% [28,29].

There were some limitations in our study. First, since flexible bronchoscopy was not available for children in our institution, it could not be used to confirm the diagnosis in suspicious cases. Second, since the city where our hospital is located experienced a major earthquake disaster and also because our hospital is located in a border region, some patients from neighbouring countries and earthquake victims did not come for postoperative check-ups, and information about these patients could not be obtained.

## 5. Conclusions

Positive clinical diagnosis of FBA is often difficult because of the low sensitivity and specificity of penetration syndrome and pulmonary auscultation. Therefore, complementary radiological examinations play an important role in reducing the rate of negative rigid bronchoscopy in children with suspected bronchial foreign bodies. Rigid bronchoscopy should always be performed as a first-line procedure in the presence of a radiopaque, obstructive foreign body on chest radiography or in the presence of characteristic clinical and radiographic signs. In doubtful cases, the presence of a foreign body should be confirmed by CT. Low-dose chest CT is a highly effective imaging modality with high sensitivity and specificity for the diagnosis of FBA in children. Since it can be performed rapidly with minimal radiation exposure and can prevent unnecessary bronchoscopies, it should be used in suspicious cases.

## Figures and Tables

**Scheme 1 tomography-11-00017-sch001:**
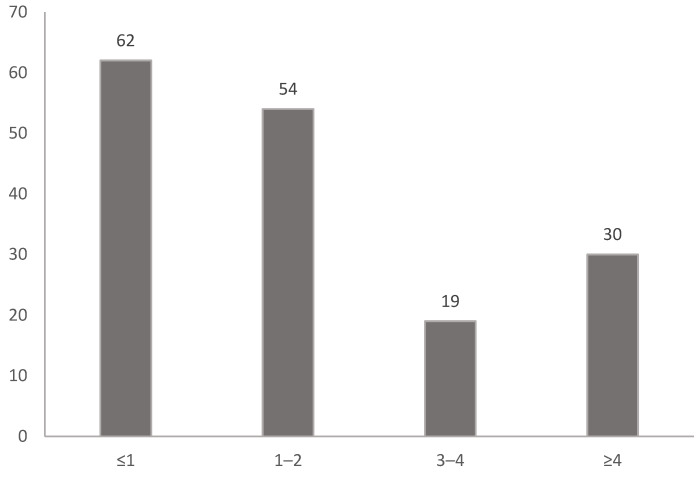
Distribution of cases by age groups.

**Scheme 2 tomography-11-00017-sch002:**
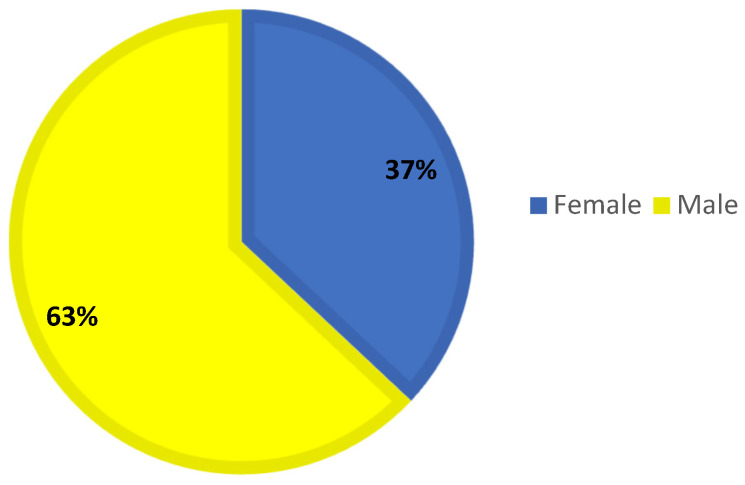
Gender distribution of cases.

**Figure 1 tomography-11-00017-f001:**
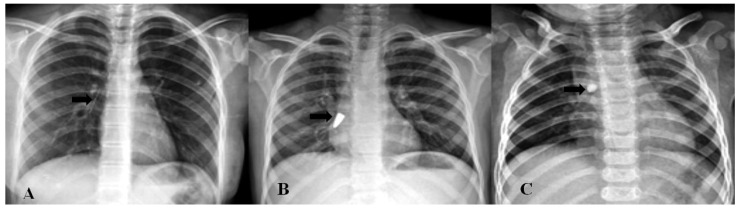
Radiopaque foreign bodies: (**A**) needle, (**B**) pen tip, and (**C**) stone (black arrows indicate foreign bodies).

**Figure 2 tomography-11-00017-f002:**
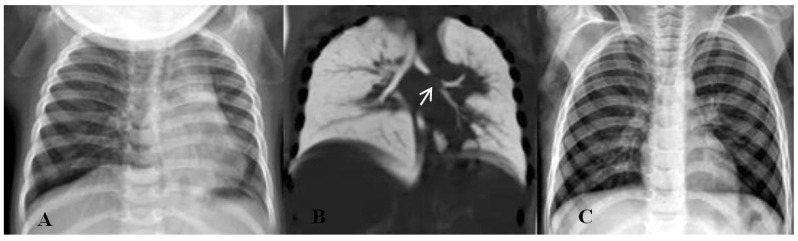
(**A**) Atelectasis obliterating the left heart contour. (**B**) Negative CT image, density in favour of a foreign body that almost completely obliterates the airway (white arrow). (**C**) Normal chest radiography after bronchoscopy.

**Figure 3 tomography-11-00017-f003:**
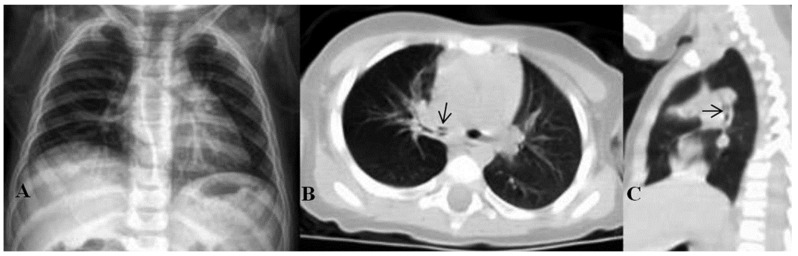
(**A**) Normal chest X-ray. (**B**) Foreign body in the right main bronchus (black arrow) in the axial CT image (although it is not observed in normal radiographs, hyperaeration in the right lung is noticeable on CT). (**C**) Foreign body in sagittal CT image.

**Table 1 tomography-11-00017-t001:** Baseline characteristics.

Sex	n (%)
Female	98 (59.4)
Male	67 (40.6)
Age median (IQR)	2 (1–3)
History	
Cough	118 (71.5)
Tachypnoea	34 (20.6)
Dyspnoea	21 (12.7)
Decreased breath sounds	5 (3)
Fever	3 (1.8)
Stridor/wheeze	5 (3)
Choking	2 (1.2)
Radiography	n (%)
FBA +	47 (28.5)
FBA −	118 (71.5)
CT	n (%)
Was taken	43 (26.1)
Was not taken	122 (73.9)
CT	n (%)
FBA +	40 (93)
FBA −	3 (74)
CT (Location of Foreign Body)	n (%)
Trachea	2 (5)
Right	20 (50)
Right and left	1 (2.5)
Left	17 (42.5)
Bronchoscopy	n (%)
FBA +	135 (81.8)
FBA −	30 (18.2)
Bronchoscopy (Location of Foreign Body)	n (%)
Trachea	12 (8.8)
Carina	6 (4.4)
Right main bronchus	12 (8.8)
Right bronchus intermedius	18(13.3)
Right and left bronchi	5 (3.7)
Left main bronchus	47 (34.9)
Complications	
Bleeding	5 (3)
Bronchospasm	8 (5)
Desaturation	47 (34.9)
Bradycardia	2 (1.2)
Cardiac arrest	1 (0.6)
Bronchoscopy—only n (%)	122 (73.9)
CT ± bronchoscopy	43 (26.1)

IQR: Q1–Q3 (n = 165).

**Table 2 tomography-11-00017-t002:** Distribution of removed foreign bodies.

Foreign Body	n (%)
Peanut	40 (29.2)
Sunflower seed	18 (13.14)
Needle	16 (11.68)
Hazelnut	7 (5.11)
Walnut	6 (4.38)
Almond	5 (3.65)
Pumpkin seeds	5 (3.65)
Carrot	4 (2.92)
Stone	4 (2.92)
Watermelon seeds	3 (2.19)
Sweetcorn	3 (2.19)
Ground peanut	3 (2.19)
Pea	2 (1.46)
Piece of meat	2 (1.46)
Pen tip	2 (1.46)
Haricot bean	2 (1.46)
Potatoes	2 (1.46)
Broad beans	1 (0.73)
Bulgur wheat	1 (0.73)
Apple	1 (0.73)
Cashew	1 (0.73)
Pen cap	1 (0.73)
Chestnut	1 (0.73)
Roasted chickpea	1 (0.73)
Metal	1 (0.73)
Nylon	1 (0.73)
Chickpeas	1 (0.73)
Fastener	1 (0.73)
Lentils	1 (0.73)
Rosary bead	1 (0.73)

**Table 3 tomography-11-00017-t003:** Compatibility and performance of CT and radiography devices with bronchoscopy being the gold standard for detecting FBA.

	Bronchoscopy							
	FBA + (n = 135) n (%)	FBA − (n = 30) n (%)	Total (n = 165) n (%)	Sensitivity	Specificity	Accuracy	κ (95%CI)	*p*
CT				%100	%100	%100	1 (1–1)	<0.001
FBA +	40 (100)	0 (0)	40 (93)					
FBA −	0 (0)	3 (100)	3 (7)					
Radiography				%34.1	%96.7	%45.4	0.144 (0.10–0.26)	0.001
FBA +	46 (34.1)	1 (3.3)	47 (28.5)					
FBA −	89 (65.9)	29 (96.7)	117 (71.5)					

CI: confidence interval, κ: kappa.

**Table 4 tomography-11-00017-t004:** Comparing groups in terms of FBA diagnosis.

	Group I (n = 122)	Group II (n = 43)	Total	*p*
	n (%)	n (%)	n (%)	
FBA +	95 (77.9)	40 (93)	135 (81.8)	0.027
FBA −	27 (22.1)	3 (7)	30 (18.2)	

Group I: bronchoscopy-only, Group II: CT ± bronchoscopy; *p* value was obtained from the Pearson Chi-square test.

## Data Availability

Data are available from the corresponding author upon request.

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
