# Peer review of "Negative Bronchoscopy or Computed Tomography Radiation in Children with Suspected Foreign Body Aspiration? Pros and Cons"

_tomography, 2025, doi:10.3390/tomography11020017_

Round 1

Reviewer 1 Report

Comments and Suggestions for Authors

- I suggest revising the conclusion of the abstract to explicitly highlight why CT is effective, particularly in the context of its role in diagnosing FBA.

- The introduction could better emphasize the core gap addressed by the study. I recommend clarifying the specific challenges or limitations in the current diagnostic approach for FBA and clearly outlining how the use of CT addresses these issues.

- Consider linking the patients' demographic data table to the Materials and Methods section.

- Is it possible to present the age distribution of patients more clearly, such as through a graph or histogram? This visual representation could enhance the reader's understanding of the study population.

- Is "direct graphy" a commonly used term in this context? If not, I suggest replacing it with a more widely recognized term, such as "direct radiography," to avoid confusion.

Comments on the Quality of English Language

The quality of English in the manuscript can be improved

Author Response

Comments 1- I suggest revising the conclusion of the abstract to explicitly highlight why CT is effective, particularly in the context of its role in diagnosing FBA.

Response 1: ''Low-dose chest CT is a highly effective and reliable imaging method for the diagnosis of FBA due to its rapid performation, minimal radiation exposure, has high sensitivity and specificity, therefore it can prevent unnecessary bronchoscopies in suspicious cases and increase the positive bronchoscopy rate.''

 Comments 2: The introduction could better emphasize the core gap addressed by the study. I recommend clarifying the specific challenges or limitations in the current diagnostic approach for FBA and clearly outlining how the use of CT addresses these issues.

Response 2: ''Up to 40% of children may have no physical findings that would suggest a foreign body and the children with foreign bodies could have normal X-rays. Small foreign bodies can remain in the respiratory tract for weeks without symptoms, and the development of pneumonia may be the first sign of aspiration. If the air can freely passes the foreign body especially in the early times after aspiration, auscultatory changes may not be detected. Although radiography is the main tool used to detect the presence of foreign bodies in the respiratory tract, it is insufficient for diagnosis, especially in the early stages, because time is required for atelectasis or emphysema to be observed.''

 Comments 3: Consider linking the patients' demographic data table to the Materials and Methods section. - Is it possible to present the age distribution of patients more clearly, such as through a graph or histogram? This visual representation could enhance the reader's understanding of the study population

Response 3: The changes you mentioned have been made. Scheme 1 and scheme 2 have been added to the material method section.

Comments 4. - Is "direct graphy" a commonly used term in this context? If not, I suggest replacing it with a more widely recognized term, such as "direct radiography," to avoid confusion.

Response 4: The changes you mentioned have been made.

Reviewer 2 Report

Comments and Suggestions for Authors

This paper explores the potential of CT compared to radiography in the diagnosis of foreign body aspiration in children.

 Overall, the article is well-structured but needs some corrections and content updates:

Abstract: Background:

Foreign Body Aspiration (FDA) - put acronym

2.1. Screening Protocol

more data on the protocol (acquisition parameters; reconstruction/reformatting parameters)

The "x-ray" term appears along the text, but must be replaced by radiography (line 132; 144; 160; table 1; 165; table 3; 172; 176; 181)

Please cite the table 3 in the text;

Line 174: "The agreement between CT with the bronchoscopy device 174 was at a statistically significant 0.144."; replace "CT" with "radiography"

Line 185-187: "...93%, whereas in Group I (only Bronchoscopy), it was significantly higher at 77.9% compared to 77.9%.(Figure 3) Additionally, it was observed that the negative diagnosis rate in Group II was significantly higher compared to Group I (Table 5)."

Check the values according to table 4.

Table 5 is equal to table 4.

Author Response

This paper explores the potential of CT compared to radiography in the diagnosis of foreign body aspiration in children.

 Overall, the article is well-structured but needs some corrections and content updates:

Comments 1: Abstract: Background:

Foreign Body Aspiration (FDA) - put acronym

Response 1: The changes you mentioned have been made in abstract section.

Comments 2: 2.1. Screening Protocol

more data on the protocol (acquisition parameters; reconstruction/reformatting parameters)

Response 2: The changes you mentioned have been made in screening protocol section.

‘’CT scans were performed on a 64-slice Toshiba Aquilion unit (Toshiba Medical Sys-tem Corporation, Otawara-Shi, Japan, Model TSX101A, 5 mm slice thickness). Axial images were acquired in supine position (feet first) from the lung’s apex to the costophrenic angle. CT scans were performed at low dose (30 mA, 80 kV) for the paediatric age group and without the use of any contrast agent. A lung window image sequence with a slice thickness of 1,25 mm was used for measurements. The coronal and sagittal multiplanar reformation was done at 3 mm thickness. Data set was transferred and stored in DICOM format on the hospital PACS system. Parents with crying or moving children wore lead aprons to keep the child still during the CT scan.However, if intense artifacts appeared in the screen images of moving children, they were excluded from the study.’’

Comments 3: The "x-ray" term appears along the text, but must be replaced by radiography (line 132; 144; 160; table 1; 165; table 3; 172; 176; 181)

Response 3: The changes you mentioned have been made.

Comments 4: Please cite the table 3 in the text;

Response 4: The changes you mentioned have been made.

Comments 5: Line 174: "The agreement between CT with the bronchoscopy device 174 was at a statistically significant 0.144."; replace "CT" with "radiography"

Response 5:

 The changes you mentioned have been made.

 Comments 6:  Line 185-187: "...93%, whereas in Group I (only Bronchoscopy), it was significantly higher at 77.9% compared to 77.9%.(Figure 3) Additionally, it was observed that the negative diagnosis rate in Group II was significantly higher compared to Group I (Table 5)."

Check the values according to table 4.

Response 5:  The changes you mentioned have been made.

‘’It was determined that there were statistically significant differences between groups in detecting FBA (p=0.027). In Group II (CT ± Bronchoscopy), the detection rate of FBA was 93%, whereas in Group I (only Bronchoscopy)the detection rate of FBA was 77.9%, this difference was statistically significant.’’

Comments 7: Table 5 is equal to table 4.

Response 7: It was noticed that table 5 was added by mistake, it was removed from the article.

Reviewer 3 Report

Comments and Suggestions for Authors

It is known that CT is very accurate in assessing the presence of foreign bodies, especially compared to radiography. However, the text did not specify whether fluoroscopy can increase the specificity of radiography. Overall, the manuscript does not add new concepts and takes up concepts from other scientific articles.

Author Response

Comments 3: It is known that CT is very accurate in assessing the presence of foreign bodies, especially compared to radiography. However, the text did not specify whether fluoroscopy can increase the specificity of radiography. Overall, the manuscript does not add new concepts and takes up concepts from other scientific articles.

Response 3: In the discussion section of the article, literature datas regarding the use of fluoroscopy in foreign body aspirations were reviewed.

‘’The use of fluoroscopy in the diagnosis of foreign body aspiration is very limited; it can only be useful in demonstrating mediastinal displacement or decreased dia-phragmatic movements on the side of aspiration. Fluoroscopy is a dynamic study that shows the ventilation of the lungs. However, it is difficult to ensure cooperation during inspiration and expiration in children. It is also a user-dependent examination and false-negative rates reach up to 50%.’’

Round 2

Reviewer 3 Report

Comments and Suggestions for Authors

See the comment concerning the reference

Author Response

Comment: See the comment concerning the reference

(Does the introduction provide sufficient background and include all relevant references? X not applicable)

Response: All references have been rearranged, doi numbers of the references have been added, and are indicated in square brackets within the article.